# Super-geometric electron focusing on the hexagonal Fermi surface of PdCoO$_2$

Maja D. Bachmann[1,2,7]*, Aaron L. Sharpe [3,4,7], Arthur W. Barnard[5], Carsten Putzke[1,6], Markus König[1], Seunghyun Khim[1], David Goldhaber-Gordon[4,5], Andrew P. Mackenzie [1,2] & Philip J.W. Moll [1,6]*

Geometric electron optics may be implemented in solids when electron transport is ballistic on the length scale of a device. Currently, this is realized mainly in 2D materials characterized by circular Fermi surfaces. Here we demonstrate that the nearly perfectly hexagonal Fermi surface of PdCoO$_2$ gives rise to highly directional ballistic transport. We probe this directional ballistic regime in a single crystal of PdCoO$_2$ by use of focused ion beam (FIB) micro-machining, defining crystalline ballistic circuits with features as small as 250 nm. The peculiar hexagonal Fermi surface naturally leads to enhanced electron self-focusing effects in a magnetic field compared to circular Fermi surfaces. This super-geometric focusing can be quantitatively predicted for arbitrary device geometry, based on the hexagonal cyclotron orbits appearing in this material. These results suggest a novel class of ballistic electronic devices exploiting the unique transport characteristics of strongly faceted Fermi surfaces.

[1] Max Planck Institute for Chemical Physics of Solids, 01187 Dresden, Germany. [2] School of Physics and Astronomy, University of St Andrews, St Andrews KY16 9SS, UK. [3] Department of Applied Physics, Stanford University, Stanford, CA 94305, USA. [4] SLAC National Accelerator Laboratory, Menlo Park, CA 94025, USA. [5] Department of Physics, Stanford University, Stanford, CA 94305, USA. [6] Institute of Materials, École Polytechnique Fédéral de Lausanne, 1015 Lausanne, Switzerland. [7] These authors contributed equally: Maja D. Bachmann, Aaron L. Sharpe. *email: majabac@stanford.edu; philip.moll@epfl.ch

Electronic conduction in metals is typically well captured by Ohm's law as frequent collisions of the electrons lead to diffusive motion and locally defined conductivity. An essential prerequisite to this common transport regime is a momentum-relaxing mean-free path, $\lambda$, that is much smaller than the size of the conductor. In extremely clean metals, however, $\lambda$ may exceed the size of the sample, and a ballistic description of charge transport becomes appropriate. The diffusive motion underlying Ohm's law is replaced by ballistic trajectories, akin to the motion of billiard balls. In such situations, geometric electron-beam optics should be achievable in solid-state devices. Significant progress has been made in two-dimensional electron systems (2DESs), where essential elements of electron optics (familiar from a free-space context) have been demonstrated in high-purity semiconducting heterostructures and graphene. These include collimated electron sources[1,2], lenses[3–5], waveguides[6,7], beam splitters[8], transistors[9], refractive[10,11], and reflective[12,13] elements. Electronic solid-state devices based on high carrier density metals differ from free-space electron-beam applications, as they operate in a quantum regime of a Fermi gas at temperatures far below their Fermi energy. Therefore, the accessible electron states for conduction are locked to the Fermi energy and the Fermi momentum, unlike the free-space electronic beams where all energies and momenta are accessible.

To date, such device concepts rely on straight-line electron trajectory segments, or in a magnetic field, circular orbits associated with small, isotropic Fermi surfaces (FS) as in graphene or 2DESs. In principle, unlike the free electron case, a solid offers the opportunity to engineer dispersion relations $E(k)$ via Bragg scattering off the lattice. By tuning the hopping integrals, the shape of the FS can be made strongly non-circular defining preferred directions of electronic motion in the solid. Here, we report a striking directionality of ballistic electron motion in the material $PdCoO_2$, arising from its almost perfectly hexagonal FS that defines three preferred directions of motion.

The metallic delafossite $PdCoO_2$ is an extraordinarily clean conductive oxide that exhibits a mean-free path of ~20 μm at low temperatures[14], rendering the delafossite metals class the most conductive oxides known[15–18]. The quasi-2D crystal consists of highly conductive, triangular coordinated palladium sheets separated by layers of insulating $CoO_2$ octahedra. Only a single, half-filled band crosses the Fermi level, resulting in a large, cylindrical FS consisting of a nearly perfectly hexagonal cross section ($k_F^{min} = 0.90$ Å$^{-1}$, $k_F^{max} = 1.00$ Å$^{-1}$)[14], which is weakly warped along $k_z$[14]. Unusual transport characteristics, such as strong momentum-conserving scattering processes have recently attracted attention, as they have been argued to lead to hydrodynamic transport[19].

Here we focus on the exotic ballistic regime of $PdCoO_2$ at low temperature (2 K) and in a range of applied out-of-plane magnetic fields ($B \leq 14$ T), arising from transverse electron focusing (TEF) in combination with the hexagonal Fermi surface. In particular, we demonstrate how this peculiar Fermi surface shape leads to unusual TEF trajectories owing to the physical fact that mostly three discrete directions of group velocity are available to the quasiparticles. The large, flat sections of a hexagon are shown to significantly enhance the TEF signals, a situation we refer to as *super-geometric focusing*. In addition, we explore how the strong deviations of the Fermi surface from a circle lead to highly crystal direction-dependent TEF.

## Results

**Transverse electron focusing (TEF) geometry**. Two narrow contacts resembling nozzles connecting along the same crystal edge and two far away large electrodes form the electric connections to the device (Fig. 1a). A uniform magnetic field $B$ is applied perpendicular to the surface, causing electrons to follow cyclotron orbits. If the nozzle separation $L$ is smaller than or comparable with the shortest microscopic mean-free path in the problem, one expects undisturbed cyclotron motion between the nozzles. Similar to the circular trajectories of electrons in free-space, electrons in a lattice move on orbits defined by the geometry of their FS. When an integer multiple $n$ of the cyclotron diameter matches the contact spacing $L$, an excess of electrons arrives at the distant contact, leading to an increase in its chemical potential. As the cyclotron radius, $r_c = \hbar k_F/eB$, is inversely proportional to the magnetic field $B$, and electrons have a finite probability to be reflected specularly back into the channel, a linearly spaced train of peaks in measured potential can be observed at fields $B_n = \frac{2n\,\hbar k_F}{eL}$, where $\hbar$ is the reduced Planck constant, $k_F$ is the Fermi vector, and $e$ is the electronic charge. These correspond to trajectories with $n–1$ bounces off the surface. First observed and studied in elemental conductors (Bi[20], Sb[21], W[22], Cu[22], Ag[23], Zn[24], and Al[25]), the TEF effect was further investigated in 2DESs[12] and graphene[26,27] and has recently been employed to spatially separate and detect electron spins in a spin–orbit coupled system[28].

**The role of the hexagonal Fermi surface**. In $PdCoO_2$, the TEF signal strongly deviates from that expected for free electrons due to the hexagonal shape of the FS. In the presence of a magnetic field, the charge carriers revolve around the FS in $k$-space in a plane perpendicular to the applied field. In real space, the shape of the cyclotron motion is given by a 90° rotation of the $k$-space orbit. In materials with circular FSs, for instance single-layer graphene or 2DESs, the electron trajectories are circular and their velocity distribution is isotropic in real space (see Fig. 1b). Interesting deviations from circular Fermi surface shapes can be observed in bi- and tri-layer graphene exhibiting trigonal warping[26], as well as graphene superlattices[27]. A peculiar situation emerges in such strongly faceted FSs as in $PdCoO_2$, in which a macroscopic proportion of states all move in the same direction as the group velocity is locally perpendicular to the FS. This large weight of parallel moving states leads to a strong geometric enhancement of the focusing capabilities of a material, which results in a highly anisotropic velocity distribution in the palladium planes with three preferred directions of motion.

On a circular FS, a simple geometric model can be used to understand the shape of the TEF peak. An electron injected at an angle $\theta$ away from normal incidence will be focused at a distance $x = 2r_c \cos\theta$ away from the nozzle. By assuming an isotropic angular distribution of electron injection, we can find the distribution of distances from the injection nozzle at which electrons return to the edge, by calculating the classical probability density function, which is given by $n(x) = \frac{2}{\pi}\frac{1}{\sqrt{(2r_c)^2 - x^2}}$ (see Supplementary Note 10 for derivation). The divergence at $x = 2r_c$ describes the focusing effect on a circular FS; even when electrons are injected evenly in all directions, those entering the device under a small angle $\theta$ will all be focused onto nearly the same spot by a magnetic field (purple shaded region in Fig. 1c). This focusing occurs due to the presence of a well-defined Fermi surface in the metal, and is hard to achieve in free-space electron beams as it depends on highly monochromatic electrons.

The hexagonal FS of $PdCoO_2$, however, hosts large flat sheets along which electrons are naturally focused onto the same point (orange shaded region in Fig. 1c). Such flat faces lead to strong deviations of the electron trajectory compared to curved ones. While the magnetic field constantly transfers momentum to the

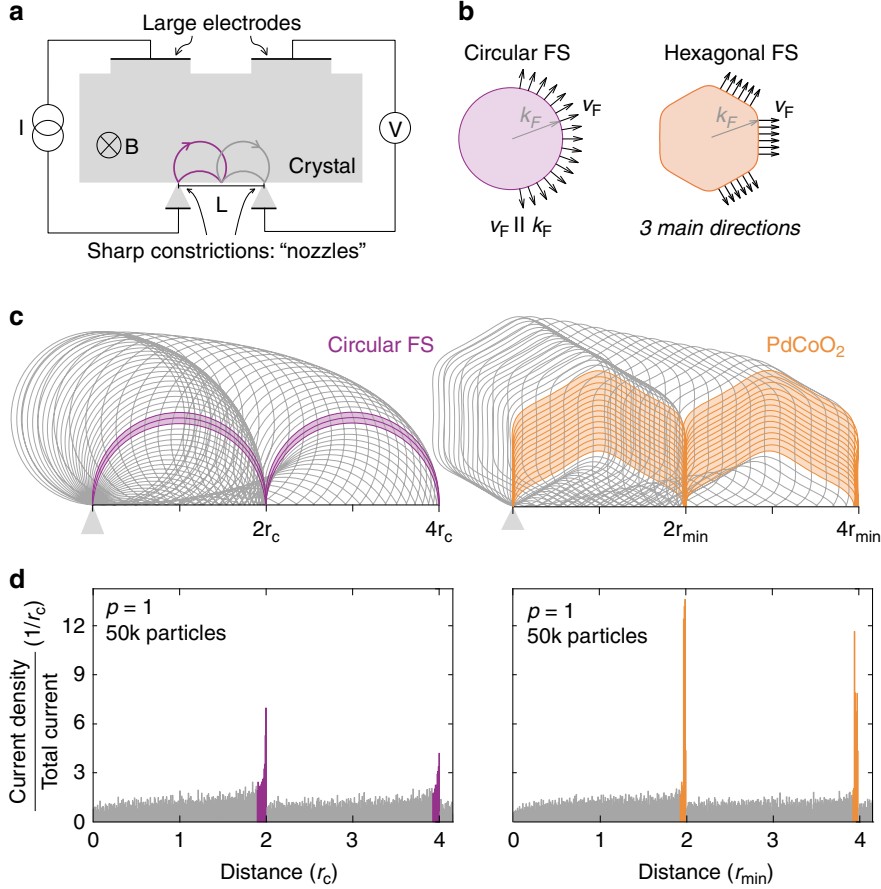

**Fig. 1** Transverse electron focusing on a circular versus a nearly perfectly hexagonal Fermi surface. **a** Experimental schematic for transverse electron focusing (TEF). Injected through a narrow nozzle, electrons in a 2D system follow an in-plane cyclotron motion when subject to an out-of-plane magnetic field. When a second, receiver nozzle is located at a distance L away from the injection nozzle, the focusing condition is met when the nozzle separation L corresponds to an integer multiple of the cyclotron diameter ($L = n2r_c$). This leads to a focusing spectrum with characteristic voltage peaks at field values corresponding to $B_n = \frac{2n\hbar k_F}{eL}$, where $n$ is a positive integer. **b** Directional restriction of the Fermi velocity $v_F$ on a nearly hexagonal FS in contrast to a circular FS. In the latter case, $v_F$ is always parallel to $k_F$ and can take any direction in real space. On a hexagonal FS, however, the large flat sections lead to a collimation of electrons into only three main directions. **c** Semiclassical trajectories of electrons injected isotropically at the origin for a circular (left) and nearly hexagonal FS (right). The trajectories contained in the shaded region give rise to a peak in the focusing spectrum in the panels below. **d** Simulated focusing spectra for a circular (left) and hexagonal FS (right) assuming completely specular boundaries ($p = 1$). The flat sides of the hexagonal FS lead to a significant increase of the focusing peak height, purely due its geometrical shape

quasiparticle changing its state in $k$-space, the group velocity in real space remains constant. This leads to a macroscopic number of states being focused onto the same spot, and we refer to this situation as super-geometric focusing to set it apart from the geometric focusing of curved Fermi surfaces. This intuitive picture of enhanced focusing is corroborated by numerical TEF simulations on the experimentally determined hexagonal FS of PdCoO$_2$ (Fig. 1d, see Supplementary Fig. 11 and Supplementary Note 11 for details on the simulations).

**Focused ion beam microstructured single crystals**. To test this prediction experimentally, we have fabricated TEF devices from as-grown single crystals of PdCoO$_2$, which grow as ultraclean single crystals without the need for any further purification. The synthesis and characterization are described in Supplementary Note 1 and Supplementary Fig. 1, respectively, as well as elsewhere[29]. The crystals grow as hexagonal platelets (~10–20 µm thick, and several 100 µm in lateral dimensions), with the growth edges of the crystal oriented 90° away from the crystal axes (Fig. 2a).

Critical for the observation of TEF as outlined above is the use of narrow injection nozzles. In two-dimensional systems,

point-like contacts can be easily defined lithographically. Here we employ a focused ion beam (FIB)-based technique to fabricate narrow nozzles of diameters as small as 250 nm ($\ll \lambda$, $L$). As a maskless technique capable of etching materials in three dimensions, FIB machining offers a unique way to carve a well-defined microsample out of a larger crystal. The details of FIB micromachining are described in Supplementary Note 2 and Supplementary Fig. 2, and can be found elsewhere[30]. Given the high degree of control over the material on the sub-µm scale, this approach may be a viable route towards a more quantitative fermiology based on TEF on even the smallest metallic samples[27].

In our typical PdCoO$_2$ devices, two sets of nozzles (Fig. 2c, bottom edge: 1–8 and left side: A–D) connect to a central square area and are arranged perpendicular to each other. The orientation of the FS and the Brillouin zone are overlaid in the middle panel. Due to the sixfold rotational symmetry of the FS, these two sets of nozzles will probe TEF directed along different parts of the FS. The bottom panel shows an enlarged view of the nozzles, which are ~250 nm wide and are separated by 1 µm each. Given the low-voltage levels of the TEF signals, high currents of multiple mA had to be applied, and the resulting high current density in the $10^6$ A cm$^{-2}$ range naturally raises questions about

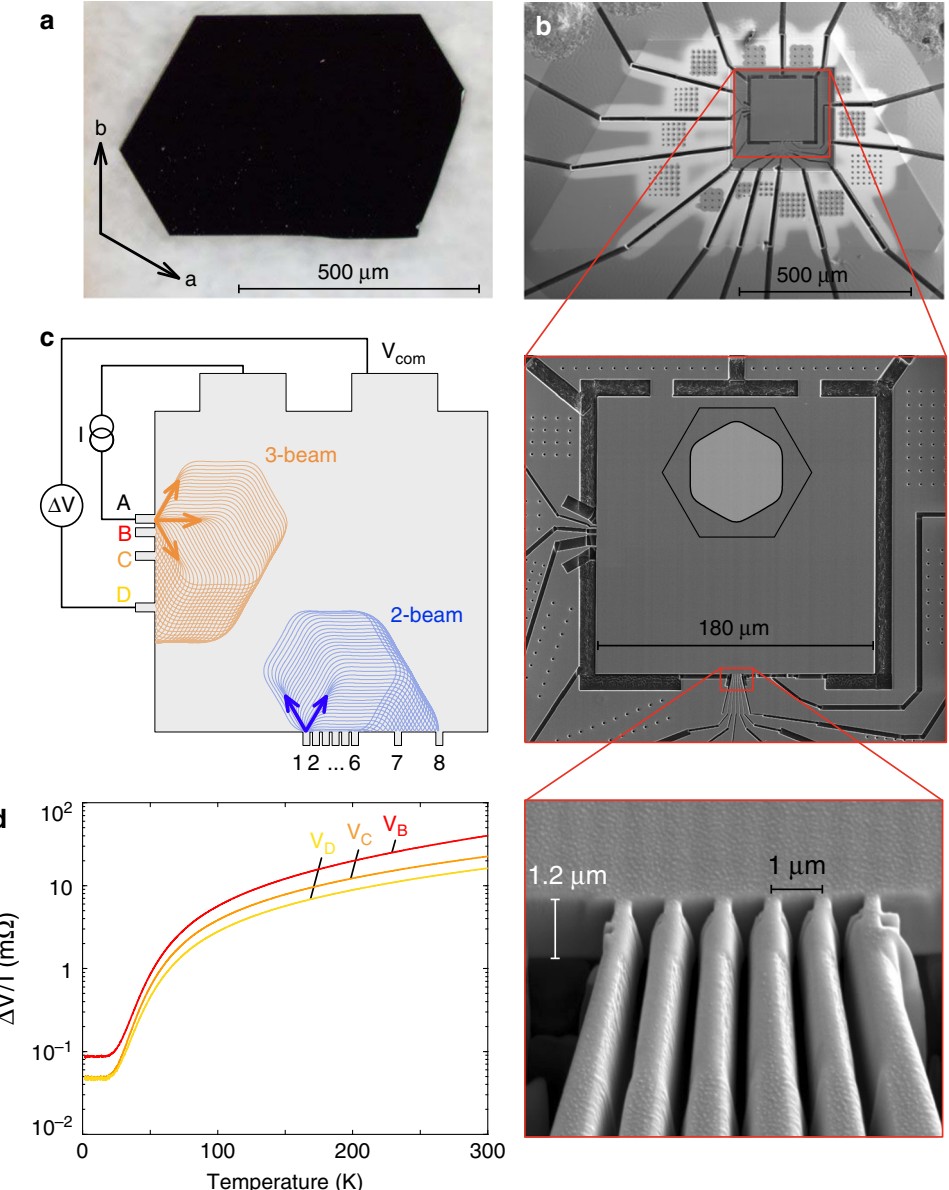

**Fig. 2** Focused ion beam defined TEF device, measurement scheme, and characterization. **a** Optical microscope image of a single crystal of PdCoO$_2$ with a thickness of ~20 μm. The crystal axes are rotated 90° away from the natural growth edges. **b** Top: scanning electron micrograph of a FIB-defined TEF device. The crystal has been top-contacted with gold and structured into a multi-terminal transport device. Middle: magnification of the central region. By top irradiation with a gallium ion beam, the crystal has been locally thinned down to ~1 μm. Two sets of nozzles, oriented 90° with respect to each other, probe TEF along the corner and the flat sides of the hexagonal FS. Bottom: side view onto the lower set of nozzles, which are separated by 1 μm and are ~250 nm wide. The long constrictions leading toward the nozzles act both as flexures to reduce fractures and as collimators. **c** Experimental setup for probing directional-dependent TEF. In all measurements, the current is sourced between the top-left large electrode and a nozzle, and the voltage is measured between the top-right electrode (V$_{com}$) and a second nozzle. **d** Nonlocal voltage signal divided by the sourced current as a function of temperature. The measured voltages are in good agreement with the solutions to the Laplace equation solved with finite-element simulations by assuming in-plane resistivities of $\rho_{300K} = 3$ μΩcm and $\rho_{2K} = 8$ nΩcm

heating and nonlinearities. No such effects were observed, and the signal remains linear in current over the entire studied range (see Supplementary Fig. 3 and Supplementary Note 3). The material can carry such extreme current densities owing to its low in-plane resistivity $\rho \sim 8$ nΩcm. The temperature dependence of the nonlocal voltages (measurement configuration Fig. 2c; data Fig. 2d) observed in our experiment is reproduced quantitatively by solving the Laplace equation numerically in our geometry by using bulk resistivity values[18]. The quantitative agreement at all temperatures suggests that the FIB-fabricated crystalline circuit matches the designed geometry, and the electronic properties

over most of the structure are not strongly perturbed by the FIB process.

**Direction-dependent TEF.** For a hexagonal FS, the TEF strongly depends on the crystallographic direction (Fig. 3), because the group velocity of a $k$-state is always perpendicular to the FS. When the electrons are ejected from a nozzle onto a flat side of the FS hexagon (nozzles A–D), they are collimated into three main directions; hence we term this configuration the "3 beam direction". If the nozzles are rotated by 90° with respect to the underlying FS (nozzles 1–8), the electrons then travel in a "2 beam direction"

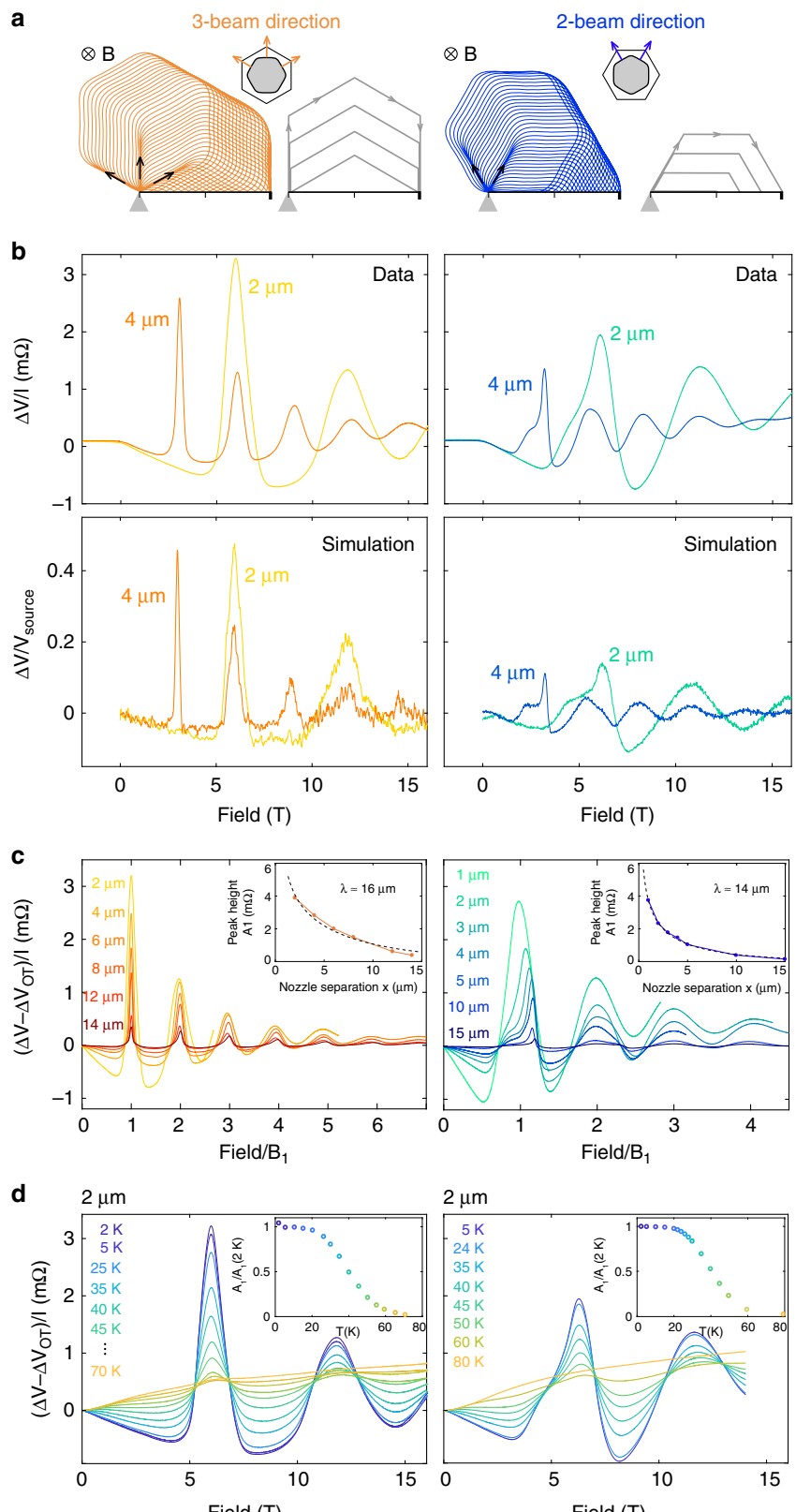

configuration. With an applied out-of-plane magnetic field, the electrons then undergo hexagonal cyclotron motion, yet with different initial conditions. Accordingly, the experimental TEF peaks from nozzle pairs of the same separation occur at different fields (see Supplementary Fig. 8, Supplementary Note 8, and Supplementary Table 2) and differ in shape between the two crystal

directions, as seen in Fig. 3b. At negative fields, there is a small, diffusive magnetoresistive component. Since there is only one type of charge carrier (electrons) present in this system, no focusing peaks are observed at negative fields. At small positive fields, where the cyclotron diameter is larger than the separation of the nozzles, a voltage inversion is observed: more electrons reach the large contact

**Fig. 3** Experimental results and ballistic simulations. **a** Comparison of the TEF geometry for nozzles (gray triangles) cut parallel to the edge of the FS (left) and the corner of the FS (right). In the former case, electrons are emitted (gray triangle) predominantly along three directions, in contrast to the 30° rotated case where only two electron jets are formed. The trajectories are sketched for perfect hexagons, and while the 3-beam direction shows ideal super-geometric focusing conditions, focusing is absent along the 2-beam direction for a hexagon with sharp corners. **b** Top row: measured voltages $\Delta V$ divided by the applied current $I = 3$ mA as a function of magnetic field for nozzle pairs separated by 2 μm and 4 μm along the 3-beam (left) and 2-beam (right) direction at 2 K. For reproduced results, see Supplementary Fig. 4, Supplementary Table 1, and Supplementary Note 4. Bottom row: ballistic simulations (no bulk scattering) for the geometries used in the measurement in the top row, with the boundary specularity set to $p = 0.1$. **c** TEF spectra for various nozzle separations, scaled such that their first peaks coincide. Inset: the height of the primary focusing peak as a function of nozzle separation, from which the mean-free path can be extracted to be about $\lambda \sim 15$ μm. **d** Temperature dependence of the TEF peaks for a nozzle pair with 2 μm separation. Inset: peak height of the primary peak scaled by its value at 2 K as a function of temperature. The decay of the peak follows the reduction in mean-free path

($V_{\mathrm{com}}$ in Fig. 2c) than the nozzle. Once the cyclotron diameter equals the distance between the nozzles a voltage peak is detected. As expected for the $B$-linear period of TEF, the second peak of the 4 μm spaced nozzles coincides with the first peak of the 2 μm spaced nozzles. The Fermi surface is encoded in the peak shape, leading to a significant enhancement of the focusing in the 3-beam compared with the 2-beam configuration. While in the 3-beam direction a large number of electrons are focused into a sharp single peak, the 2-beam direction displays a broad shoulder followed by a peak of reduced amplitude. Simple geometric arguments show that for an ideal hexagon, the 3-beam direction would exhibit a divergent super-geometric focusing, while the 2-beam direction would not focus at all (see Supplementary Figure 6 and sketch in Fig. 3a). The focusing peak in the 2-beam configuration as well as the rounding of the peak in the 3-beam data arise from the rounded corners of the real Fermi surface as compared with the mathematical hexagon. This is confirmed by Monte Carlo simulations by using a tight-binding approximation of the Fermi surface[14] based on ARPES data[31] (see Supplementary Note 11), which reproduce the observed peak shapes and their fine structure (Fig. 3b).

**TEF as a probe of the mean-free path**. TEF probes semiclassical trajectories of ballistic electrons, and in the absence of scattering would lead to focusing over arbitrary distances. In real crystals, scattering is always present, limiting the range over which focusing can be observed. Accordingly, the height of the first TEF peak shrinks as the nozzle distance is increased (Fig. 3c). It is intuitively clear that the TEF signal will be strongly suppressed for nozzles between which a typical ballistic path would be longer than the mean-free path (see Supplementary Fig. 5 and Supplementary Note 5). A more rigorous calculation, comparing with the observed exponential decay of the focusing signal with nozzle separation, allows us to extract a mean-free path on the order of $\lambda \approx 15$ μm from the data (see Supplementary Fig. 9, Supplementary Note 9, and Supplementary Table 3). This value estimated from TEF is in good agreement with previous studies estimating $\lambda$ from transport by using a simple Drude model[19]. As the temperature is increased, the normalized amplitude of the primary peak stays almost constant up to 20 K (Fig. 3d). This is consistent with a roughly constant—and long—momentum-relaxing mean-free path in this temperature region, as reflected by the temperature-independent resistivity observed self-consistently in the devices (Fig. 2d) and in measurements on macroscopic crystals[14]. From the analysis of a previous flow experiment[19], a momentum-conserving mean-free path of ~2 μm was deduced for PdCoO$_2$ <20 K. The possible effects of this were not included in the models used for the simulations presented in Fig. 3b or Supplementary Fig. S7. Although the simulations clearly capture the main features of our observations very well, there are differences in the details, generally seen in the simulations containing sharper features than the experimental data; it is possible that these differences would be reconciled by including momentum-conserving scattering in a more complete analysis. Above 20 K,

the focusing peaks gradually decrease until they cease to exist ~70 K, presumably primarily due to the reduction of the momentum-relaxing mean-free path due to Umklapp electron–electron and electron–phonon scattering.

**Enhanced apparent specularity due to hexagonal Fermi surface**. The first peak $B_1$ plays a special role in the TEF geometry as it describes ballistic paths between both nozzles without any interaction with the device boundary. All other peaks at $B_n$, $n > 1$, correspond to paths that include $(n-1)$ scattering events on the sidewall between the nozzles. This has been exploited to use TEF to perform surface spectroscopy[24,32], based on the probabilistic nature of the surface reflection of the quasiparticle back into the channel. A scattering event may be specular, thus conserving its momentum component parallel to the surface, or diffuse, leading to a random continuation of the trajectory after the impact. The specularity of the surface, $p$, denotes the probability of specular reflection. Conventionally, the amplitude ratios between subsequent peaks, $q_n = \frac{A_{n+1}}{A_n}$, are used to estimate $p$, as each subsequent peak corresponds to a trajectory that differs by one additional surface impact[33]. However, even in a simplified analysis adopting the above assumptions, in the super-geometric focusing configuration the point-spread function is so sharply peaked at $2r_c$ (see Fig. 1d) that electrons will be statistically refocused onto the next peak, regardless of the specularity. In agreement, we observe peaks up to $n = 8$ in the devices, yielding a $q$-value of roughly 0.6 (Fig. 4). Numerical simulations (see Supplementary Fig. 7 and Supplementary Note 7) corroborate this finding, showing that even barely specular surfaces ($p = 0.1$) are compatible with a $q$-value around 0.4 (Fig. 4b).

**Discussion**

Our experimental data strongly support the hexagonal shape of the semiclassical orbits in PdCoO$_2$, which impacts the TEF signatures. In particular, the directional dependence of the TEF signal as well as the strong distortion of the peak in the 2-beam configuration unequivocally evidence non-circular trajectories. The details of the signal are well reproduced by simulations of ballistic transport based on a tight-binding Hamiltonian describing the realistic hexagonal Fermi surface. While the essentially hexagonal shape explains the key features of super-geometric focusing, significant features appear due to the deviation from the ideal hexagon. Most prominently, the 2-beam direction is not expected to show any sharp focusing feature in the case of a perfectly hexagonal Fermi surface. Its experimental observation is entirely due to the rounding of the corners, which is well captured by calculations, thus confirming that the present models well describe even subtle details of its electronic system. Besides the trajectories in the bulk, the hexagonal Fermi surface shape will also alter the collimating properties of the nozzles and influence their capability to select certain $k$-states, which will be the topic of further studies on the TEF dependence on the nozzle

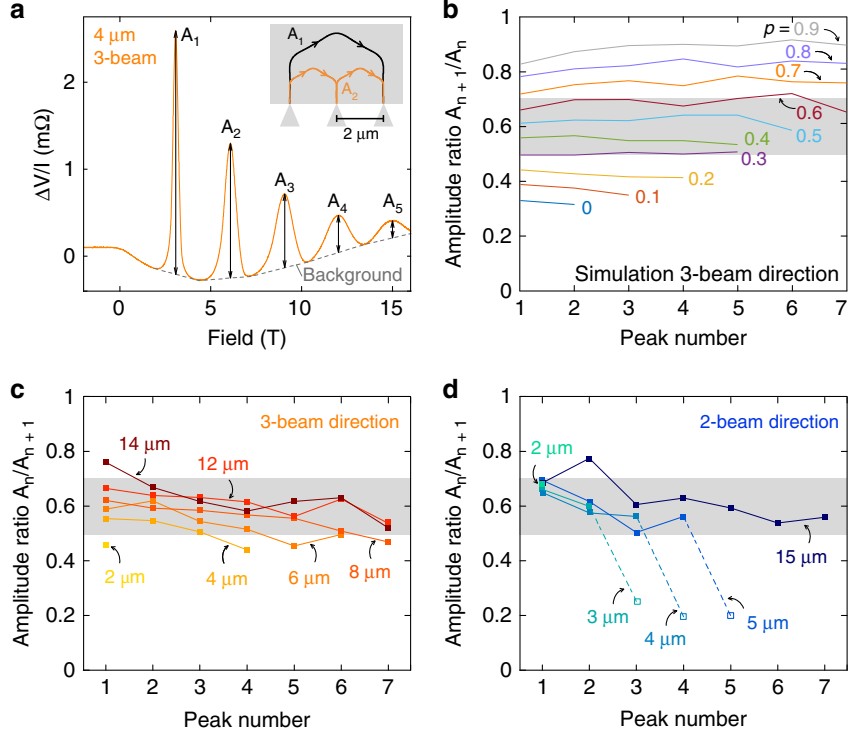

**Fig. 4** Specularity of FIB-defined boundaries. **a** Definition of the peak amplitudes after subtraction of a smooth background that is obtained by connecting the voltage minima. The inset shows the trajectories giving rise to the first and second focusing peaks with amplitudes $A_1$ and $A_2$. Although the second focusing peak directly interacts with the boundary where there is another nozzle, its amplitude $A_2$ is not affected. This again reflects the insensitivity of the focusing of trajectories on the surface specularity in super-geometric focusing conditions. **b** Simulated amplitude ratios for a range of boundary specularity values $p$ between 0 and 1 in a purely ballistic model without bulk scattering. For small $p$-values, the amplitude ratio $q$ is strongly enhanced due to super-geometric focusing. **c** Amplitude ratio $q = \frac{A_n}{A_{n+1}}$ for various nozzle pairs measured along the 3-beam direction. The $q$-factor remains at a constant value of roughly $q = 0.6 \pm 0.1$ for all higher harmonic focusing events. **d** Along the 2-beam direction, the extracted $q$-factor takes a similar value of $q = 0.6 \pm 0.1$ as in the 3-beam direction. When the cyclotron diameter becomes smaller than the nozzle width $b \approx 0.3\,\mu m$ at fields greater than $B = \frac{\hbar k_F}{2eb} \approx 11.3$ T, the peak amplitude rapidly decreases in size (dashed lines, empty squares) and deviates from a power-law behavior

width and length. In addition, these results suggest the feasibility of novel electronic devices operating in the ballistic limit exploiting strong deviations from circular Fermi surfaces. This additional avenue of control will enable novel types of functionality. For example, selectively aligning parts of a $PdCoO_2$ crystalline circuit along the 3-beam or the 2-beam direction will completely alter the ballistic response of the device, despite it being chemically and structurally homogeneous. Intriguingly, $PdCoO_2$ is an extremely conductive ballistic metal, and therefore automatically incorporates low dissipation, a key prerequisite for high-power and high-frequency applications that is unlikely to be achievable in low carrier density devices based on graphene and 2DES. Promising recent thin-film results[34] may indicate a pathway toward larger-scale fabrication of such devices.

The almost perfectly hexagonal FS shape of $PdCoO_2$ arises from accidental fine-tuning of hopping parameters in its band dispersion. In general, super-geometric focusing is a generic property of materials with large parallel sections on their FS. Such flat areas on Fermi surfaces can be engineered in 2D materials where arbitrary control over the chemical potential and sometimes even the band structure is possible via gating, as has been shown in bilayer graphene[26] or in graphene-based moiré super-lattices[27]. Alternatively, flat Fermi surface sections are not rare in bulk crystals such as $PdCoO_2$, and future material science efforts may uncover ballistic behavior in other ultraclean metals. Our FIB-based approach showcases a viable route toward the investigation of ballistic behavior in challenging materials, where the crystal size or chemical composition may impede traditional

lithography-based methods to fabricate ballistic devices on the sub-μm scale.

## Data availability

All data needed to evaluate the conclusions in the paper are present in the paper. All raw data underpinning this publication can be accessed in comprehensible ASCII format at DOI: https://doi.org/10.5281/zenodo.3406604. Source code for the simulations can be found at https://github.com/dgglab.

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

## Acknowledgements

The project was supported by the Max-Planck Society and has received funding from the European Research Council (ERC) under the European Union's Horizon 2020 research and innovation programme (grant agreement No. 715730). M.D.B. acknowledges studentship funding from EPSRC under grant no. EP/I007002/1. A.L.S. acknowledges support from a Ford Foundation Predoctoral Fellowship and a National Science Foundation Graduate Research Fellowship. A.L.S. would like to thank Edwin Huang for helpful discussions and Tom Devereaux for letting us use his group cluster. Computational work was performed on the Sherlock cluster at Stanford University and on resources of the National Energy Research Scientific Computing Center, supported by DOE under contract DE_AC02-05CH11231. D.G.G.'s and A.W.B.'s work was supported by the U.S. Department of Energy, Office of Science, Basic EnergySciences, Materials Sciences and Engineering Division, under Contract No. DE-AC02-76SF00515.

## Author contributions

M.D.B., C.P., M.K., and P.J.W.M. fabricated the microstructures, and M.D.B., C.P., and P.J.W.M. performed the measurements. The crystals were grown by S.K. and A.P.M., A.L.S., A.W.B., and D.G.G. performed the ballistic simulations. All authors were involved in the design of the experiment and writing of the paper.

## Competing interests

The authors declare no competing interests.
