## [Peer Review File · Nature Communications]

Reviewers' comments:

Reviewer #1 (Remarks to the Author):

The paper by Bachmann et al. is an experimental work supported by numerical simulations reporting the effect of super-geometric electron focusing in the metallic oxide (PdCoO₂) with hexagonal Fermi surface. Transverse electron focusing in a perpendicular magnetic field has proven to be a powerful method to investigate ballistic electron dynamics in isotropic, or nearly isotropic metals, semiconductors, and recently extensively used in graphene. Super-geometric focusing results from strongly anisotropic Fermi surfaces selecting preferred directions of electron beam propagation in single crystalline materials.

To the best of my knowledge this experiment is a première, which is made possible by the recent availability of high-quality delafossite crystals hosting a quasi-hexagonal Fermi surface. Sample geometry and nano-fabrication are excellent with a high-mobility macroscopic sample cleverly equipped with two sets of nano injectors and probes oriented along high symmetry edges exhibiting contrasted magnetic focusing patterns. The measurement is supported by comprehensive numerical simulations enabling to account for non-idealities. Some conclusions of the experiment can be regarded as direct evidences of the super-focusing effect, others rely on the support of simulations. Altogether the experiment is beautiful and very convincing and deserves publication in Nature Communications. If the paper is well written and documented by illustrative figures, I still consider that the presentation can be improved according to the lines discussed below and that some issues need to be clarified.

The paper contains a large body of experimental and theoretical results. Some are direct signatures of the super-geometric focusing (overall anisotropy of TEF, large amplitude of the first peak, its specific peak shape, reduction of the focusing field, robustness to multiple edge reflections). Other, like the non-vanishing TEF in the 2-beam configuration or the quantitative analysis of the multiple reflection peaks amplitude ratio, rely on a detailed comparison with simulations taking into account deviations from ideality: the low specularly of edge reflection in metals, the finite size of the probing nozzles, the warping of the hexagons etc. In this respect, important information is spread over the main text and the method section, sometimes hidden in the captions of the supplementary figures. The readability of the paper would benefit from a recapping in a concluding paragraph of the direct and indirect (simulation dependent) signatures of super-geometric focusing, as well as its limitations. In particular, the observation of a reduced focusing field in the super-geometric case should be discussed. I could estimate a factor 0.92 between 2-beam and 3-beam from inspection of Fig.3B, which is not quite the factor 0.87 of the ideal hexagons, but eventually consistent with the warped hexagon case in Fig.S4C. Please also correct Fig.3A where three beam focusing is inconsistently sketched at $2r_c$.

Few more comments (main text):

The abstract sentence « The peculiar hexagonal Fermi surface naturally leads to electron self-focusing effects in a magnetic field, well below the geometric limit associated with a circular Fermi surface ». It is unclear to me what means "below".

The list of geometrical optics experiments in the introduction might be updated by including the ballistic graphene transistor by K. Wang et al. (PNAS 2019) or the corner reflector by H. Graef et al. (arXiv:1901.02225v1).

In the 4th paragraph, it would be useful to give the value of the Fermi wavelength in PdCoO₂, especially in the perspective of the discussion on edge scattering.

It is not straightforward for the reader to identify which set of nozzles in Fig2 correspond to 2-beam and 3-beam focusing in Fig3. In paragraph 11 : please specify the nozzle-series (A-D) after the sentence « When the electrons are injected flat zone of the FS hexagon,... » and (1-8) after « if the nozzles are rotated ... »

In paragraph 13 and Figs4 and S5, should it be that the peak amplitude ration is $A_{(n+1)}/A_n$ instead of $A_{(n)}/A_{(n+1)}$?

In paragraph 13 there is a missing list of Refs on line 4 : [refs].

Few more comments (Methods):

Some of the method sections reduce to a mere figure and caption. The reader would deserve a few commenting sentences. For example, S2 present another sample which is presumably equipped with 2-beam nozzles only. To discuss reproducibility issues, it would be interesting to know if measurement in this sample (or the other samples) is consistent with the data shown in the main text.

Electron focusing is expected to be a linear response property. The experiment uses quite large current injections $\sim 24\ 000\text{A/m}$ with a current of 6mA (Fig.S3) in a nozzle of 250nm. If I can understand that the voltage resolution requires large currents in such a very low resistance sample, I am concerned about the linearity issue. One could easily figure out that large currents are needed to ensure a uniform current density in such an anisotropic material, but then the question arises on the robustness of the focusing with inhomogeneous current densities. In addition, one could be concerned about self-heating effects at nozzle injection, or eventually electron-electron interaction effects at the injectors, such as considered in Ref. 27. A discussion of these issues should be included in the paper (main text or method section).

Reviewer #2 (Remarks to the Author):

The manuscript is well written and presents important results in the field of ballistic transport in engineered Fermi Surface material such as PdCoO₂. The transverse magnetic focusing scheme used clearly demonstrates that a hexagonal Fermi surface manifests in anisotropic motion of electrons under magnetic field. As opposed to the traditional peak in magnetic focusing non-local voltage at the contact a cyclotron diameter away, a peak in voltage is observed at a lower spacing. It would be nice to confirm that this change is not merely due to a lower effective mass of electrons in the material. In other words, how would the results be different if only the effective mass of electrons were different in a circular Fermi surface? I would suggest the authors to put in the numbers for the effective mass of electrons. The results are novel for the community in ballistic electronics especially for electron optics experiments. With the above mentioned revision, I highly recommend publication of the manuscript.

Reviewer #3 (Remarks to the Author):

The manuscript "Super-geometric electron focusing on the hexagonal Fermi surface of PdCoO₂" by Bachmann et al. is on the topic of transverse electron focusing (TEF) on PdCoO₂, a material with a hexagonal Fermi surface (FS). Such FS gives rise to highly directional ballistic transport in contrast with more commonly studied materials with isotropic FS. Specifically, with an applied out-of-plane magnetic field, these systems present a TEF with hexagonal cyclotron motion where focusing depends on the crystallographic orientation with respect to the sample axis.

The subject matter of this work is interesting. However, I find some relevant issues (listed below) that should be addressed before I can recommend this manuscript for publication.

1- Despite TEF experiments have been mainly realized in materials with circular (isotropic) FS, there are TEF studies undertaken in materials with non-circular FS, too. This should be explained in the introduction to facilitate the reading and understanding of the manuscript. Examples are bilayer or trilayer graphene with trigonal warping [Taychatanapat et al. Nature Phys. 2013, Ref 26 in the manuscript] or graphene superlattices [Lee et al. Science 2016, Ref. 27 in the manuscript]. Authors briefly mention these two studies in their conclusions, however, this information should appear earlier in the text.

2- I think authors should be cautious when using the terms “super-geometric electron focusing” and “geometric limit”, such terminology may confuse the reader.

The prefix “super” is commonly used when overcoming a clear fundamental physical limit (see for instance [Krisna Kumar et al. Nat. Physics, 13, 1182 (2017)]).

In the present study, however, there are several cases where hexagonal FS do not lead to higher focusing peaks than circular FS. For instance, hexagonal FS along the 2-beam direction (Fig. S4). Furthermore, I note that the “geometric (focusing) limit associated with a circular FS” does not seem to be a well-defined limit: such focusing limit is extremely dependent on the collimation of the injected electron beam. For instance, a fully collimated beam injected into a TEF device with circular FS will produce focusing peaks with the same heights as the hexagonal FS along the 3-beam direction.

As such, I find confusing the title and key sentences in the abstract such as “The peculiar hexagonal Fermi surface naturally leads to electron self-focusing effects in a magnetic field, well below the geometric limit associated with a circular Fermi surface.” or “This super-geometric focusing can be ...”. I recommend avoiding these generic expressions.

3- Apart from the terminology used, the present study does not demonstrate an enhanced focusing on the hexagonal FS of PdCoO₂ in the 3-beam direction with respect to materials with circular FS and isotropic electron injection as indicated in the title, abstract and introduction. Fig.1 predicts such enhanced focusing, but no material with circular FS is measured. Authors should accordingly modify these three parts of the manuscript.

4- Regarding the focusing data shown along 3-beam or 2-beam directions and their corresponding simulations.

- Authors should analyze and compare further the main measured and simulated focusing peaks (Fig. 3b). For instance, magnetic fields at which focusing peaks occur for 3-beam and 2-beam configurations should be stated. Such fields are important since they should be consistent with the different cyclotron diameters exhibited by the two configurations (Fig. S4). Also, peak height ratios between 3-beam and 2-beam directions in experimental data and simulations seem relevant. According to Fig. S4... could this ratio be used to extract the real warping of the hexagonal FS in these experiments?

- Above a temperature $T = 20\text{K}$, focusing peaks gradually decrease due to the reduction of the momentum-relaxing mean free path in the system. Could authors determine further whether this is due to Umklapp electron-electron or electron-phonon scattering? For instance, a T^2 dependence of the resistivity would point towards electron-electron scattering [Lee et al. Science 2016, Ref. 27 in the manuscript; Lucas et al. Phys.Rev.B, 97, 045105 (2018)]. This information is important and useful to compare the behaviour observed in these devices with respect to other systems reported in literature [Lee et al. Science 2016, Ref. 27 in the manuscript].

5- Other minor observations:

- Page 5. Some references are missing (“[refs]” is written instead).

- Page 25. Authors state “The simulations of Fig. 2B of the main text are comprised of 1001 magnetic field points, each consisting of 30000 charge carriers...”. Figure 2b are SEM images, it seems that authors are referring to other figure in the text.

Dear Reviewers,

We thank you for your time and efforts to review our manuscript "Super-geometric electron focusing on the hexagonal Fermi surface of PdCoO₂". We highly appreciate your encouraging comments about the novelty of these experiments. In our view, this paper reports both a new approach to electron beam manipulation in complex materials by FIB-engineering of crystalline geometries, as well as unusual electron trajectories arising from the highly planar Fermi surface segments. These are quite rare as such materials become usually unstable against density waves due to their strong nesting, and here PdCoO₂ can provide novel transport features.

Following your suggestions, we have performed additional measurements to pinpoint the geometric focussing properties due to the unusual Fermi surface shape of PdCoO₂. In particular, we have established the absence of non-linearities in current, which is non-trivial given the small size of the signals; as well as the nozzle dependence of the focussing peaks. These new data, combined with an improved analysis, support the main findings even stronger.

In the following, we address your comments point-by-point:

Reviewer 1

1. The paper contains a large body of experimental and theoretical results. Some are direct signatures of the super-geometric focusing (overall anisotropy of TEF, large amplitude of the first peak, its specific peak shape, reduction of the focusing field, robustness to multiple edge reflections). Other, like the non-vanishing TEF in the 2-beam configuration or the quantitative analysis of the multiple reflection peaks amplitude ratio, rely on a detailed comparison with simulations taking into account deviations from ideality: the low specularly of edge reflection in metals, the finite size of the probing nozzles, the warping of the hexagons etc. In this respect, important information is spread over the main text and the method section, sometimes hidden in the captions of the supplementary figures. The readability of the paper would benefit from a recapping in a concluding paragraph of the direct and indirect (simulation dependent) signatures of super-geometric focusing, as well as its limitations.

We have added such a summary statement to the paper. This wrap up is indeed helpful to guide the reader and to also help them to identify the future challenges arising from this work.

2. In particular, the observation of a reduced focusing field in the super-geometric case should be discussed. I could estimate a factor 0.92 between 2-beam and 3-beam from inspection of Fig.3B, which not quite the factor 0.87 of the ideal hexagons, but eventually consistent with the warped hexagon case in Fig.S4C. Please also correct Fig.3A where three beam focusing is inconsistently sketched at $2r_c$.

These observations are completely correct. We have added a full analysis of the focusing fields. Indeed the expected ratio is 0.92 and the deviation from the ideal hexagon is due to the rounding of the corners. Experimentally, the ratio is even higher, around 0.96-0.99. While close, the deviations from the simulations may indicate imperfections in the nozzle transfer function (deviations from homogeneous k-state injection). Our next project concerns the investigation of the focusing peak position and shape on the size of the nozzles. Both their width and length will influence the collimation expected from them. We also added a statement on this in the concluding paragraph.

We have changed the Figure accordingly. The idea was here to scale the figure to the nozzle distance, defining an "effective" r_c . This way the peaks overlap directly and thus guide the reader more towards the differences in their functional form. We agree though that this is more confusing than helpful, as a key point of this work is that there simply is no r_c . Now the peaks do not coincide in the x-axis anymore as they should, however this makes it a bit harder to directly compare their shapes.

3. The abstract sentence « The peculiar hexagonal Fermi surface naturally leads to electron self-focusing effects in a magnetic field, well below the geometric limit associated with a circular Fermi surface ». It is unclear to me what means "below".

This sentence was not ideally phrased and has been changed to "The peculiar hexagonal Fermi surface naturally leads to enhanced electron self-focusing effects in a magnetic field compared to circular Fermi surfaces"

4. The list of geometrical optics experiments in the introduction might be updated by including the ballistic graphene transistor by K. Wang et al. (PNAS 2019) or the corner reflector by H. Graef et al. (arXiv:1901.02225v1).

Thank you for pointing these recent developments out, we added these papers accordingly.

5. In the 4th paragraph, it would be useful to give the value of the Fermi wavelength in PdCoO₂, especially in the perspective of the discussion on edge scattering.

Akin to the problem of r_c , on the strongly non-circular Fermi surface there is also no well defined k_F . We have added the minimal and maximal values as determined by the comparison of quantum oscillations and ARPES to ab-initio calculations. For the question of surface scattering, it is very clear though that all possible k-states are half-Brillouin zone sized and hence atomic in wavelength. This is explicitly stated.

6. It is not straightforward for the reader to identify which set of nozzles in Fig2 correspond to 2-beam and 3-beam focusing in Fig3. In paragraph 11 : please specify the nozzle-series (A-D) after the sentence « When the electrons are injected flat zone of the FS hexagon,... » and (1-8) after « if the nozzles are rotated ... »

We have labelled the directions in Fig. 2C and specified the nozzles in the text accordingly.

7. In paragraph 13 and Figs4 and S5, should it be that the peak amplitude ration is $A_{(n+1)}/A_n$ instead of $A_{(n)}/A_{(n+1)}$?
8. In paragraph 13 there is a missing list of Refs on line 4 : [refs]

Yes, thank you. Both has been fixed.

9. Some of the method sections reduce to a mere figure and caption. The reader would deserve a few commenting sentences.

Indeed this was too short to be helpful. We have substantiated the text in the supplement.

- a. For example, S2 present another sample which is presumably equipped with 2-beam nozzles only. To discuss reproducibility issues, it would be interesting to know if measurement in this sample (or the other samples) is consistent with the data shown in the main text.

This is correct. We have added a new section on reproducibility. We now provide additional data in further devices that completely reproduce the findings of the paper.

- b. Electron focusing is expected to be a linear response property. The experiment uses quite large current injections $\sim 24\ 000\text{A/m}$ with a current of 6mA (Fig.S3) in a nozzle of 250nm . If I can understand that the voltage resolution requires large currents in such a very low resistance sample, I am concerned about the linearity issue. One could easily figure out that large currents are needed to ensure a uniform current density in such an anisotropic material, but then the question arises on the robustness of the focusing with inhomogeneous current densities. In addition, one could be concerned about self-heating effects at nozzle injection, or eventually electron-electron interaction effects at the injectors, such as considered in Ref. 27. A discussion of these issues should be included in the paper (main text or method section)

Thank you for raising this important point. You also pinpoint the challenge to observe the voltages at lower current levels. We have performed extensive checks for non-linearity, and observed none. First of all, we did not detect any higher harmonic signals in the lock-in amplification. Then we probed this more directly by recording the same trace at different ac-current levels. Indeed we can vary the current by almost 2 orders of magnitude, without any observations of non-linearities. Given how well our data is resembled by the simulations, which are computed in the linear response approximation, this is consistent, but it is definitely critical to confirm experimentally. This important information is given in the text, and in a more extended discussion in the supplement. The current density around 1MA cm^{-2} is indeed extremely high, and the experimental fact that not a single

nozzle thus far has been damaged due to electrical breakdown speaks for the high conductivity and high purity of PdCoO₂ crystals.

Reviewer 2

1. As opposed to the traditional peak in magnetic focusing non-local voltage at the contact a cyclotron diameter away, a peak in voltage is observed at a lower spacing. It would be nice to confirm that this change is not merely due to a lower effective mass of electrons in the material. In other words, how would the results be different if only the effective mass of electrons were different in a circular Fermi surface? I would suggest the authors to put in the numbers for the effective mass of electrons.

Thank you for this suggestion. We have stated the effective mass observed by quantum oscillations, as well as our own quantum oscillation analysis in the manuscript. The effective mass is around $1.7m_e$ and hence we do not expect any deviations due to the effective mass.

To first order, the effective mass does not enter this problem as unlike in free space the velocity of the quasiparticle is fixed to the Fermi velocity. The cyclotron radius, $r_c = m v_F / (qB)$, can be expressed in terms of the Fermi wavevector, $\hbar k_F / (2\pi m) = v_F$, thus eliminating the effective mass as $r_c = \hbar k_F / (2\pi qB)$. Hence the TEF radius is a purely geometric property of the Fermi surface, irrespective of small changes of effective mass. This is of course only true within linear response theory, and if strong electric fields accelerate the quasiparticles away from the Fermi surface sufficiently, indeed a deviation would be possible.

We have tested this by performing a current dependence now shown in the supplement and explained in the main text. Our results clearly show the system to be in the linear response regime, thus we can exclude that the effective mass tricks us here.

Reviewer 3

1. Despite TEF experiments have been mainly realized in materials with circular (isotropic) FS, there are TEF studies undertaken in materials with non-circular FS, too. This should be explained in the introduction to facilitate the reading and understanding of the manuscript. Examples are bilayer or trilayer graphene with trigonal warping [Taychatanapat et al. Nature Phys. 2013, Ref 26 in the manuscript] or graphene superlattices [Lee et al. Science 2016, Ref. 27 in the manuscript]. Authors briefly mention these two studies in their conclusions; however, this information should appear earlier in the text.

Thank you for this comment. We are sorry if we left the impression to not reference these works properly, this was not our intention. Our work is quite distinct from them, as the ballistic anisotropy was not the focus of these studies which we directly probe through the direction dependence of the nozzles. We now mention these studies earlier as suggested.

2. I think authors should be cautious when using the terms "super-geometric electron focusing" and "geometric limit", such terminology may confuse the reader. The prefix "super" is commonly used when overcoming a clear fundamental physical limit (see for instance [Krisna Kumar et al. Nat. Physics, 13, 1182 (2017)]). In the present study, however, there are several cases where hexagonal FS do not lead to higher focusing peaks than circular FS. For instance, hexagonal FS along the 2-beam direction (Fig. S4). Furthermore, I note that the "geometric (focusing) limit associated with a circular FS" does not seem to be a well-defined limit: such focusing limit is extremely dependent on the collimation of the injected electron beam. For instance, a fully collimated beam injected into a TEF device with circular FS will produce focusing peaks with the same heights as the hexagonal FS along the 3-beam direction. As such, I find confusing the title and key sentences in the abstract such as "The peculiar hexagonal Fermi surface naturally leads to electron self-focusing effects in a magnetic field, well below the geometric limit associated with a circular Fermi surface." or "This super-geometric focusing can be ...". I recommend avoiding these generic expressions.

The use of terminology is very specific to different fields as well as tastes of researchers. It does not appear to reflect the use in the field to link the prefix "super" to overcoming fundamental physical limits in literature. There are cases in which this is true, as the mentioned "super"-ballistic transport or for example optical "super"-resolution, but many more where this is not the case. It simply means "above" or "beyond", and is used in scientific literature as such, for example in the commonly used in general terms "super"-linear,

“super“-cell/“super“-structure. Countless times this suffix is used in our meaning, to describe something going beyond or above something else, such as “Super-tough carbon-nanotube fibers” [A.B. Dalton et al., Nature 423, 703 (2003)] or “super high power mid-infrared femtosecond light bullet” [P. Panagiotopoulos et al., Nature Photonics 9, 543-548 (2015)].

Of course, terminology is not relevant but the key point is to help the reader understand and interpret the work accordingly. We mean by “super-geometric focusing” the enhancement of TEF due to the presence of flat sections of the Fermi surface, which increases geometrically the transfer function between nozzles and that is absent on circular or ellipsoidal Fermi surfaces. We strongly feel that coining this name substantially enhances the reading flow of the manuscript compared to often describing this situation repetitively.

Accordingly, we hope to convey our reasoning for this and that you find the improved description of this term in the abstract as well as the main text clearer. We fully agree to your concerns on the use of “geometric limit”. Here the use of “limit” was clearly incorrect, and it should have read “case of circular Fermi surface”. We have completely rephrased this to eliminate this confusing point.

3. Apart from the terminology used, the present study does not demonstrate an enhanced focusing on the hexagonal FS of PdCoO₂ in the 3-beam direction with respect to materials with circular FS and isotropic electron injection as indicated in the title, abstract and introduction. Fig.1 predicts such enhanced focusing, but no material with circular FS is measured. Authors should accordingly modify these three parts of the manuscript.

Indeed no material with circular Fermi surface was measured, however the direct comparison of different materials in TEF is very difficult. Even when only the first peak without interface conditions is considered, the density and detailed nature of the scattering centers enter into the shape of the peaks (distribution, density, types, scattering cross-sections, ratio of small-angle vs. large-angle scattering, etc). This is impossible to replicate in another material.

The key point of our work is both the enhancement and the orientation dependence of the peaks. The enhancement of focusing over that of a hypothetical material with circular Fermi surface follows from straightforward geometric considerations. In all devices, we see a strong difference in peak amplitude at identical nozzle distance between the 2-beam and the 3-beam direction. The peak height is a measure of the number of modes that connect the two nozzles at a given field. The 2-beam and 3-beam directions are the extrema of this function. The 2-beam direction has the smallest and the 3-beam direction the largest number of trajectories, while all other possible orientations lie in between. If the charge carrier density of the hypothetical circular material is the same, it is clear that the number of channels must be the geometric mean between these two peaks. In other words, if the circular material shows the same peak height as the 3-beam direction here, and it necessarily would do so for all crystal orientations as the FS is circular, then the total number of trajectories is significantly larger.

4. Regarding the focusing data shown along 3-beam or 2-beam directions and their corresponding simulations.
- Authors should analyze and compare further the main measured and simulated focusing peaks (Fig. 3b). For instance, magnetic fields at which focusing peaks occur for 3-beam and 2-beam configurations should be stated. Such fields are important since they should be consistent with the different cyclotron diameters exhibited by the two configurations (Fig. S4). Also, peak height ratios between 3-beam and 2-beam directions in experimental data and simulations seem relevant. According to Fig. S4... could this ratio be used to extract the real warping of the hexagonal FS in these experiments?

Thank you for raising this point. Indeed the focusing fields and height ratios provide important information on the electronic system and these were missing. We now have a detailed discussion and state all values in a table in the supplement. If the editor agrees on the length, we are more than happy to move this into the main manuscript as it provides important information.

Regarding the real warping of the Fermi surface, we are less optimistic though that this can be done at a precision exceeding the results of ARPES and quantum oscillations, which yield a Fermi surface that already well explains the peaks. Now one needs to consider the Fourier coefficients of the Fermi surface as fitting parameters and optimize the peak ratio to the experimental value. The main issue here is the large mathematical dependences between these fitting parameters, as many terms influence the peak in similar ways. We currently aim to do a slight modification of your suggestion. Using a high precision rotator, we

aim to identify the detailed warping by fitting angular magnetoresistance oscillations. Then using the detailed parameters established in our sample, we aim to recompute TEF peaks for different nozzle sets in such a way that the signal shows a clear signature of the correct warping terms. For example, rounding of the main faces of the hexagon leads to a bifurcation of the central current beam, which in the right conditions should be observable as a splitting of the main TEF peak. Given the long time it takes to make such a device from a crystal, this model-driven approach will probably lead better results in the determination of the Fermi surface.

- Above a temperature $T=20\text{K}$, focusing peaks gradually decrease due to the reduction of the momentum-relaxing mean free path in the system. Could authors determine further whether this is due to Umklapp electron-electron or electron-phonon scattering? For instance, a T^2 dependence of the resistivity would point towards electron-electron scattering [Lee et al. Science 2016, Ref. 27 in the manuscript; Lucas et al. Phys.Rev.B, 97, 045105 (2018)]. This information is important and useful to compare the behaviour observed in these devices with respect to other systems reported in literature [Lee et al. Science 2016, Ref. 27 in the manuscript].

This is an excellent observation, and was one of our initial working hypothesis to start investigating the TEF. In particular, we were interested in the strength of momentum-conserving scattering to investigate the hydrodynamic electron transport regime we proposed in an earlier work [P. Moll et al., Science 351, 1061 (2016)]. Naturally, irrespective of the conservation of electron momentum, any scattering event removes the particle from the trajectory and hence TEF is reduced by both scattering types alike. It turns out, however, that the situation is not as simple as we believed. The key point is that the assumption of an isotropic scattering rate τ is strongly violated and a non-trivial k -dependence of $\tau(k)$ has to be taken into account. This is to explain the significant differences between the mean-free-path estimated from a Dingle analysis of quantum oscillations, $\lambda \sim 0.6\mu\text{m}$ [C.W. Hicks et al., Phys. Rev. Lett. 109, 116401 (2012)], and the much longer transport and TEF mean-free-path of $\lambda \sim 25\mu\text{m}$. Initially, this was suggested to arise from phase smearing due to field inhomogeneities, yet our results on μm -sized FIB-cut transport bars confirm this Dingle analysis even when the sample is too small to expect notable inhomogeneities. Our current hypothesis is a strongly enhanced scattering cross-section at the corners of the hexagon, which remains to be tested.

Regarding the aspect of the functional form of the T -dependence of the resistivity, this is very hard to quantify in the TEF devices. Owing to the high in-plane conductivity, the voltage drop across the device is very small, thus precluding an accurate determination of resistivity in this structure. We have aimed to estimate the consistency of our devices with data measured in proper resistivity studies [C.W. Hicks et al.]. The voltage profile in the devices was computed using their resistivity and solving Laplace equation, see Fig. 2d. Indeed we quantitatively agree with their results. However, we cannot improve their estimates of resistivity, as ours holds significantly larger systematic error bounds due to the unfavorable geometry for resistivity measurements.

5. - Page 5. Some references are missing ("[refs]" is written instead).
- Page 25. Authors state "The simulations of Fig. 2B of the main text are comprised of 1001 magnetic field points, each consisting of 30000 charge carriers...". Figure 2b are SEM images, it seems that authors are referring to other figure in the text.

Thank you for pointing these mistakes out.

Reviewers' comments:

Reviewer #1 (Remarks to the Author):

Congratulations for this beautiful piece of work. I have no reservation to recommend publication in Nature Communications

Reviewer #2 (Remarks to the Author):

The manuscript has been revised to reflect changes suggested by the reviewers. The paper is well written and the results are novel enough to be published. The findings of the paper support their claim of observation of super-geometric electron focusing on a hexagonal Fermi surface of PdCoO₂. I recommend publication of the manuscript as is.

Reviewer #3 (Remarks to the Author):

This document is a revision of a manuscript I initially reviewed. My report included a number of key points (requests for clarification and technical remarks) to be addressed. I've now read the authors' rebuttal letter to each of the points I raised and the new version of the manuscript.

Most of these points have been satisfactorily clarified. Still, I think that an important discussion about the existing mismatch between experiments and theory is missing in the main text.

In more detail:

I think authors have properly addressed points 1, 2 and 5. Key references are introduced and described in the initial part of the main text. Also, the meaning of "super-geometrical focusing" is clearer now in the abstract/introduction.

Regarding point 3. I understand the explanation given by authors for an ideal system. My concern here was more "experimental". Due to sample inhomogeneities, peak ratios between different nozzles (separated by the same distance) other than 1 (around 1-2) could occur even in materials with circular FS. However, I see that distorted peaks are not expected to occur in this case then.

Regarding point 4. According to simulations (Fig.3b), a clear and consistent demonstration of differences in focusing along 3-beam and 2-beam directions would comprise: i) specific peak-shapes in these 2 directions, ii) the ratio of focusing fields between 2-beam and 3-beam being ~ 0.87 and iii) a peak height in the 3-beam case much larger than the peak height in the 2-beam case. These points are clearly shown even when accounting for realistic (ARPES-based) rounding of the corners of the hexagonal FS (Fig.3b): the focusing field ratio is ~ 0.9 and the ratio between peaks in the two configurations is >4 .

i) is seen experimentally. However, upon inspection of the experimental data in Fig.3b, ii) focusing fields in experiments seem similar in both cases (this is confirmed in the new table 2, part S8, with a ratio ~ 1), and iii) the peaks height of the 3-beam seem only ~ 2 times larger than the 2-beam case.

I think authors should describe and discuss these additional differences between experiments and theory. This information should appear in the main text, it is important.

Authors briefly mention in the supplementary information, S8 "A possible explanation for this is the difference in nozzle width between the 2-beam a 3-beam direction".

Is that all? In principle, these additional discrepancies could also be due to a "more circular" FS

existing in the actual device. Then, do simulations match better the experimental data – in terms of i), ii) and iii) above- if the considered FS is more circular? I note that the distortion calculated for the 2-beam direction is stronger than the one measured in experiments, too.

Referee 3

Dear referee,

Thank you for your time to review our manuscript. Please find our responses to the individual points below:

I think authors have properly addressed points 1, 2 and 5. Key references are introduced and described in the initial part of the main text. Also, the meaning of “super-geometrical focusing” is clearer now in the abstract/introduction.

Regarding point 3. I understand the explanation given by authors for an ideal system. My concern here was more “experimental”. Due to sample inhomogeneities, peak ratios between different nozzles (separated by the same distance) other than 1 (around 1-2) could occur even in materials with circular FS. However, I see that distorted peaks are not expected to occur in this case then.

Indeed, both statements are completely correct. Sample inhomogeneities could distort peak ratios, yet their shape should be unchanged.

Regarding point 4. According to simulations (Fig.3b), a clear and consistent demonstration of differences in focusing along 3-beam and 2-beam directions would comprise:

- i) specific peak-shapes in these 2 directions,
- ii) the ratio of focusing fields between 2-beam and 3-beam being ~ 0.87 .
- iii) a peak height in the 3-beam case much larger than the peak height in the 2-beam case. These points are clearly shown even when accounting for realistic (ARPES-based) rounding of the corners of the hexagonal FS (Fig.3b): the focusing field ratio is ~ 0.9 and the ratio between peaks in the two configurations is >4 . i) is seen experimentally.

However, upon inspection of the experimental data in Fig.3b, ii) focusing fields in experiments seem similar in both cases (this is confirmed in the new table 2, part S8, with a ratio ~ 1), and iii) the peaks height of the 3-beam seem only ~ 2 times larger than the 2-beam case. I think authors should describe and discuss these additional differences between experiments and theory. This information should appear in the main text, it is important.

Authors briefly mention in the supplementary information, S8 “A possible explanation for this is the difference in nozzle width between the 2-beam a 3-beam direction”. Is that all? In principle, these additional discrepancies could also be due to a “more circular” FS existing in the actual device. Then, do simulations match better the experimental data – in terms of i),ii) and iii) above- if the considered FS is more circular? I note that the distortion calculated for the 2-beam direction is stronger than the one measured in experiments, too.

Here we may not fully understand the point, please excuse if we miss it. We understood this comment as the request to further discuss the differences between the experiment and the simulation. In our view, however, we have a different assessment of the importance of the differences between the data and simulations, and argue that these minor details occur due to the simplifying geometrical assumptions in the simulations which do not capture the complexity of the device. In the following, we would like to lay out our argument.

First, we are sorry for a typo in table 2, the ratio between the simulated 2-beam and 3-beam direction is actually 0.96 and not 0.9 as stated previously.

Nozzle separation	Simulation			Experiment		
	2-beam direction	3-beam direction	ratio	2-beam direction	3-beam direction	ratio
2 μ m	6.19 T	5.96 T	0.96	6.04 T	6.00 T	0.99
4 μ m	3.23 T	2.98 T	0.92	3.18 T	3.07 T	0.965

This is already evident by eye from Figure 3, as the peaks in both cases do clearly not deviate by 10%. The differences of peak ratios hence are 3% and 4.5% respectively. Given the minor differences and the reproduction of the main features by the simulation, it leaves us with the impression that the match between theory and experiment is actually remarkably good. Please note that this is an ab-initio calculation that has not been adjusted to match the data at this level. The fact that even subtle details like the shoulders of the first peak are well captured in the simulation suggests clearly that the model has the required complexity to describe the data. Naturally, the simulations can always be improved to resemble the data more perfectly. The most commonly used approach would be phenomenological tuning parameters. Here we specifically did not take this route and instead present raw expected signals from the calculation.

We can clearly exclude a more circular Fermi surface in these devices, as ARPES was measured on crystals from the same batch and the quantum oscillations we observe in the devices themselves (S1) are identical to all published studies on this material. Conceptually, it is also difficult to envision which effect would give such a distortion. PdCoO₂ is a large bandwidth metal, and significant modifications are required to notably distort the Fermi surface. Its charge carrier density is very high (1 electron/Pd), hence significant chemical substitution would be required to meaningfully alter this. Alternatively, one might invoke a strain/pressure scenario, yet again enormous pressures would be required to modify a metal with a half-filled band appreciably. In addition, these extreme effects would need to be extraordinarily homogeneous and isotropic, as to preserve the notable mean-free-path in these crystals. Focusing over distances of 35 μ m is well observed after all (S5).

It is also important to note that the focusing field ratio closer to 1 is by no means an indication of a more circular Fermi surface. On the contrary, the realistic Fermi surface shows sign-changes in its curvature and resembles a star shaped object that is clearly further away from a circle yet its ratio is closer to one. There also is no bound to this ratio at one. If one continues this distortion into a heavily star-shaped object, arbitrarily big ratios well exceeding 1 would occur.

When it comes to the peak height ratio, we observe about a factor of 2 while the voltages in our simulations would indicate rather 4. This naturally goes beyond the predictive power of the model. The simulated geometry is shown in Figure S11 and here for your reference. The ohmic terminals are

connected to short, cone-like nozzles merging with the main body of the object. In the real device, carved out of one single crystal, naturally the ballistic region extends over the entire crystal in the mm range. The actual voltages observed at the much further away ohmic contacts are a result of the established voltage equilibrium in this complex object. With the simulations we show what to expect when the geometry is kept exactly the same, yet the crystal orientation is changed. Experimentally, this idealization is not possible of course. We have to work with as-grown crystals and each nozzle set is unique and the crystal paths behind the nozzle are strongly different. For example, Fig2b shows that the 2-beam nozzles are much longer and narrow compared to the 3-beam nozzles, which was necessary to follow the natural growth features of the crystal.

Given that this reduced model already consumed significant supercomputer time, it is unfortunately impractical to simulate the entire device.

Here we show the simulations scaled to the data by a simple multiplication (no offset), which we prefer to avoid in the manuscript. It is reassuring to see that the relative peak ratios, the decay with magnetic field, is reproduced very well by the simulations suggesting that a simple multiplication with a field-independent factor is justified. Naturally there are still differences and a more complex model can be built to match the data better, as always, yet there are no deviations at all suggesting that the conclusions of this paper are incorrect.

At the end of the day, the goal of this manuscript is to demonstrate the peculiar focusing expected from a hexagonal Fermi surface, which comprises of a significant peak height and shape difference between 90° rotated nozzle sets, which is clearly observed experimentally and is supported by data without taking the simulations into account. However, it is gratifying to see how these experimental findings are furthermore excellently reproduced by a simple model. We hope you agree that clearly mentioning these differences as we already did in the previous version improves the accessibility compared to moving this discussion into the main text.

We have completely rewritten the supplement S8 to better explain the geometric issues of the peak positions, and have corrected the mistake in Table 2. We thank you again for your very valuable input which has helped to shape the paper into its present form.

REVIEWERS' COMMENTS:

Reviewer #3 (Remarks to the Author):

Authors have corrected a typo in Table 2 and have satisfactorily answered my last questions regarding the mismatch between theory and experiment. Initially, I had suggested placing these discussions in the main text rather than in the supplementary information. However, after amending Table 2, the agreement between theory/experiment is better than the one shown in the previous version of the manuscript. As such, I do not mind that these additional (more technical) discussions remain in the supporting information.

I recommend the publication of this article as it is.